# "Classifying-together" phenomenon in fish (*Xenotoca eiseni*): Simultaneous exposure to visual stimuli impairs subsequent discrimination learning

**Valeria Anna Sovrano**[1,2]*, **Greta Baratti**[1], **Davide Potrich**[1], **Tania Rosà**[1], **Veronica Mazza**[1]

**1** Center for Mind/Brain Sciences, University of Trento, Trento, Italy, **2** Department of Psychology and Cognitive Science, University of Trento, Trento, Italy

* valeriaanna.sovrano@unitn.it

**Data Availability Statement:** All relevant data are within the paper and its Supporting Information files.

## Abstract

When animals are previously exposed to two different visual stimuli simultaneously, their learning performance at discriminating those stimuli delays: such a phenomenon is known as "classifying-together" or "Bateson effect". However, the consistency of this phenomenon has not been wholly endorsed, especially considering the evidence collected in several vertebrates. The current study addressed whether a teleost fish, *Xenotoca eiseni*, was liable to the Bateson effect. Three experiments were designed, by handling the visual stimuli (i.e., a full red disk, an amputated red disk, a red cross) and the presence of an exposure phase, before performing a discriminative learning task (Exp. 1: full red disk vs. amputated red disk; Exp. 2: full red disk vs. red cross). In the exposure phase, three conditions per pairs of training stimuli were arranged: "congruence", where fish were exposed and trained to choose the same stimulus; "wide-incongruence", where fish were exposed to one stimulus and trained to choose the other one; "narrow-incongruence", where fish were exposed to both the stimuli and trained to choose one of them. In the absence of exposure (Exp. 3), the discrimination learning task was carried out to establish a baseline performance as regards the full red disk vs. amputated red disk, and the full red disk vs. red cross. Results showed that fish ran into retardation effects at learning when trained to choose a novel stimulus with respect to the one experienced during the exposure-phase (wide-incongruence condition), as well as after being simultaneously exposed to both stimuli (narrow-incongruence condition). Furthermore, there were no facilitation effects due to the congruence compared with the baseline: in such a case, familiar stimuli did not ease the performance at learning. The study provides the first evidence about the consistency of the classifying-together effect in a fish species, further highlighting the impact of visual similarities on discrimination processes.

**Funding:** This study was funded by the intramural financing to V.A.S. and V.M. from their institution, the Center for Mind/Brain Sciences (CIMeC) of the University of Trento. V.A.S., V.M and G.B. received salary from the University of Trento. The funder had no role in study design, data collection and analysis, decision to publish, or preparation of the manuscript.

**Competing interests:** The authors have declared that no competing interests exist.

## Introduction

The "classifying-together" phenomenon [1,2] occurs when animals are simultaneously exposed to two visual stimuli belonging to the same context, and such a perceptual experience hinders the subsequent discrimination learning of those stimuli. In other words, after being exposed to two stimuli simultaneously, animals are slower in classifying the two stimuli apart. The effect has been discovered by Bateson and Chantrey [1] by assessing the learning performance of monkeys and chickens after a period of exposure to different visual objects. In this series of experiments, monkeys and chickens were allowed to familiarize themselves with pairs of stimuli (in the former case, a figure of the number 2 *versus* a figure of the number 5, or a figure of the letter "H" *versus* a figure of the letter "K", while in the latter case, blue moving cylinders *versus* red ones, or blue moving cylinders *versus* green ones), and then trained to choose one item from each pair to obtain food rewards. As a control, other groups of monkeys were not exposed to any figures, while other groups of chickens were exposed to cylinders of one color (green or red). Results showed that animals familiar with the coupled visual objects took a higher number of trials to learn the discrimination, with an accuracy of 90% compared to animals not exposed to any stimuli or familiar with one single stimulus. The effect was consistent in chickens with moving imprinting objects: in fact, chicks exposed to one of the two objects learned the discrimination tasks in fewer trials than those not exposed, while chicks exposed to both moving stimuli took more trials to reach the criterion in the subsequent discrimination learning [3]. The hypothesis by Bateson and Chantrey was that animals would learn the characteristics of their environment due to mere exposure: the features of a given object would be classified together, while those of different objects would be classified separately. This idea would explain why animals reared simultaneously with two stimuli later find them difficult to discriminate, precisely because those stimuli were perceived simultaneously and therefore classified together, as being part of the same visual unit.

This phenomenon, which could be named "Bateson effect" [2], is at odds with previous evidence by Gibson and colleagues [4,5]. This series of interesting experiments with rats, found facilitation in discriminating between two shapes (a circle and a triangle) after a prolonged phase of simultaneous exposure ($\approx$ 90 days from birth) to those stimuli. On the other hand, rats reared in the absence of any visual stimulation reached the learning criterion slower and made many more errors than those reared in the presence of such enrichment.

A following batch of experiments by Stewart and colleagues [6] tried to reproduce the Bateson effect by designing and performing a wide range of experimental situations in chicks. In their experiment, authors checked for different parameters, like the color of the stimuli, the age of animals and the period of rearing; however, they never strictly replicated the classifying-together phenomenon.

In summary, the facilitation/retardation phenomena after exposure to figural stimuli, in visual discrimination learning tasks, showed different results: Gibson and collaborators [4,5] identified facilitation effects, Bateson and Chantrey [1] inhibition effects, while Stewart and collaborators [6] found no effect.

Given the supposed inconsistency of the behavioral data collected in a small number of vertebrates (chicks, monkeys, rats), the present study aimed to investigate the classifying-together effect in redtail splitfin fish (*Xenotoca eiseni*), that is, a teleost widely employed in manifold studies on animal cognition (social laterality: [7,8]; spatial abilities: [9–16]; detour behavior: [17]; visual discrimination learning and optical illusions: [18–23]; numerical abilities: [24]), thus also extending the investigation to a class of vertebrates phylogenetically remote from birds and mammals. To this end, we performed two discrimination learning experiments after rearing different groups of fish with pairs of two-dimensional visual stimuli for 50 consecutive

days. In the present study we decided to use simple geometric shapes (also considered for visual discrimination learning and amodal completion in zebrafish [25]): a full red disk and an amputated red disk in *Experiment 1* (*Visual discrimination learning between a full red disk and an amputated red disk*), a full red disk and a red cross in *Experiment 2* (*Visual discrimination learning between a full red disk and a red cross*). In both Experiments 1 and 2, for each pair of stimuli, three conditions were arranged: "congruence", where fish were exposed and trained to choose the same stimulus; "narrow-incongruence", where fish were exposed to both training the stimuli and trained to choose one of them (the same condition present in [1]); "wide-incongruence", where fish were exposed to one stimulus and trained to choose the other one (condition not present in [1], investigated here for the first time). To determine the potential spontaneous preference of fish towards one or more stimuli, we further added *Experiment 3* (*Baseline definition*) by carrying out the same discrimination learning experiments as above but without previous exposure to those shapes.

If the simultaneous familiarization with two different stimuli slowed down the discrimination learning process, the exposed groups of fish should learn slower than the non-exposed ones in both *Experiment 1* and *Experiment 2*. In addition, if there were different levels of complexity as regards the nature of the discrimination associated with the congruence (i.e., where the rearing stimulus is the same as for the training one), wide-incongruence (i.e., where fish were exposed to one stimulus and trained to choose the other one) and narrow-incongruence (i.e., where fish were exposed to both training the stimuli and trained to choose one of them) conditions, fish could significantly spend more or less time to take a correct decision. Lastly, if fish did not show any innate preference towards particular shapes, they should get equal performance when trained to choose the full disk, the amputated disk or the cross.

## Materials and methods

### Ethics statements

The present research was carried out in the Animal Cognition and Neuroscience Laboratory (A.C.N. Lab.) of the CIMeC (Center for Mind/Brain Sciences) at the University of Trento (Italy). All husbandry and experimental procedures complied with European Legislation for the Protection of Animals used for Scientific Purposes (Directive 2010/63/EU) and were previously authorized by the University of Trento's Ethics Committee for the Experiments on Living Organisms, and by the Italian Ministry of Health (auth. num. 1111/2015-PR; 848/2020-PR).

### Animals and housing

Subjects were 112 mature male redtail splitfin fish *Xenotoca eiseni*, ranging from 3 to 5 cm in body length and coming from a breeding stock in our laboratory.

42 *X. eiseni* took part in *Experiment 1* (*Visual discrimination learning between a full red disk and an amputated red disk*), 42 fish took part in *Experiment 2* (*Visual discrimination learning between a full red disk and a red cross*), and 28 fish took part in Experiment 3. Because of high motivation to rejoin female conspecifics, only males were engaged in these experiments (following the same procedure reported in previous studies [18–25]).

Before the exposure to the experimental conditions, fish were maintained under a 14:8-h LD cycle and kept in a large glass tank of 150 l capacity. This tank was enriched with polychromatic small gravel and plants, thus cleaned through a suitable filter (Aquarium Systems Duetto 100, Newa, I) to ensure a comfortable habitat. The water temperature was maintained at 25 ±1˚C, and fish were fed twice a day with dry food (GVG-Mix, Sera GmbH, D). During the exposure, all the above maintenance guidelines were followed, except for the rearing tanks.

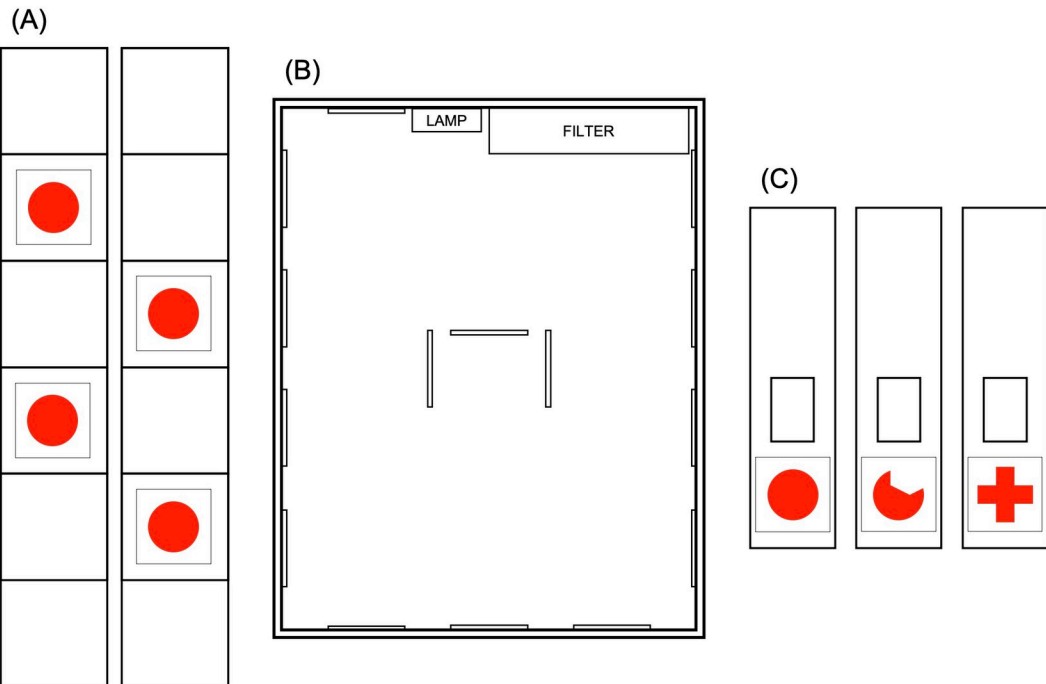

**Fig 1. Rearing panels, rearing tank, and training panels.** In (A), there is an example of the rearing panels used for the exposure-phase: The stimuli were alternatively laid out and then placed within the rearing tank. In (B), there is a schema displaying the array of the rearing panels within the rearing tank: 12 panels were equipped all along the perimeter, while 3 panels (depicting stimuli on both sides) in the center of the tank. In (C), there are the training panels used for the training-phase: The stimuli were placed beneath the corridor entrance (C).

Specifically, fish were assigned to 6 smaller glass tanks (three tanks for each experiment, measuring 35 x 30 x 28 cm) of 25 l capacity, where the two-dimensional visual stimuli were provided. Within these tanks, fish were assigned to different exposure conditions: in *Experiment 1*, 14 fish were raised in the presence of full red disks, 14 fish in the presence of amputated red disks, and 14 fish in the simultaneous presence of full red disks and amputated red disks; in *Experiment 2*, 14 fish were raised in the presence of full red disks, 14 fish in the presence of red crosses, and 14 fish in the simultaneous presence of full red disks and red crosses. For each rearing tank, 36 stimuli were glued on white plastic panels (5 x 30 cm) that were arranged along the tank's walls and in the center (see Fig 1A and 1B). The rearing tanks were externally covered with white plastic walls to isolate fish from the surrounding environment.

## Experiment 1: Visual discrimination learning between a full red disk and an amputated red disk

**Apparatus and stimuli.** In their rearing tanks, fish were exposed to the visual stimuli for 50 days, and until the end of the experiment inside individual tanks during training pauses ("exposure-phase"). The 42 animals were raised separately in three tanks (35 x 28 x 30 cm), in the presence of full red disks (N = 14), amputated red disks (N = 14) or in the presence of both (N = 14). The stimuli were full red disks ("scarlet", RGB: 255 red, 32 green, 0 blue; area: 4.52 cm$^2$, perimeter: 7.54 cm) and amputated red disks, which were missing a chunk of their upper portion ("scarlet", RGB: 255 red, 32 green, 0 blue; area: 3.22 cm$^2$, perimeter: 7.74 cm). In this experiment, the magnitude of the stimuli was balanced in order to make their perimeters as equal as possible.

After the exposure, fish performed the visual discrimination learning ("training-phase"). The experimental apparatus was the same used in previous studies investigating visual discrimination in *X. eiseni* [16–21], and it consisted of a white plastic (Poliplak®) octagonal arena (oblique segment: 4 x 15 cm; straight segment 9 x 15 cm) inscribed in a squared one (15 x 15 x 15 cm); see Fig 2A. The arena was then placed in a larger rectangular tank (57 x 18 x 38 cm) to get a surrounding comfortable region with plants, food, and two female conspecifics (not engaged in the behavioral observations), hence providing an incentive to go out of the arena for the experimental subjects [18–25]. Fish could leave the innermost zone of the apparatus by crossing two rectangular corridors (3 x 4.5 cm, 2.5 cm in length, 2.5 cm from the floor, see Fig 2B) located at the two diagonally opposite corners of the arena. At the end of each corridor, there was a door (3 x 4.5 cm) that could be easily pushed and bent by fish with their snout: the upper part of each door was a sheet of opaque plastic material (2.5 x 4.5 cm), while the lower part was a flexible sheet of transparent acetate (0.5 x 4.5 cm) (Fig 2B). The two doors were visually identical, but only one could be opened, since the other was blocked by means of a green wire clip. Also, the closed doors allowed regular water flow through their thin fissures at the edges, to avoid any potential nonvisual cues detected by extra-visual sensory modalities [14,26].

Exit attempts made by fish (i.e., choices for corridors) were clearly visible in video recordings and codified as such only in case fish entered the corridor along its whole length (2.5 cm). The waterproof stimuli (plasticized cardboard) used for the visual discrimination learning task were located below each corridor, where two transparent acetate screens (9 x 4 cm) were equipped to prevent fish from getting too close to the stimuli, thus keeping a suitable distance for an overall vision of the shapes (Fig 2A).

In Experiment 1, the stimuli to be discriminated were the same used in the three conditions of the exposure-phase within the rearing tanks: full red disks ("scarlet", RGB: 255 red, 32 green, 0 blue; area: 4.52 cm$^2$; perimeter: 7.54 cm) and amputated red disks ("scarlet", RGB: 255 red, 32 green, 0 blue; area: 3.22 cm$^2$; perimeter: 7.74); see Fig 1C.

The experimental apparatus was placed in a darkened room and was lit centrally from above (30 cm from the arena) through a white fluorescent light bulb (18 W; Osram GmbH, D). The apparatus rested on a turntable, which allowed the experimenter to rotate it (90°, conventionally clockwise) at the end of each trial, to eliminate any extra-tank cues. The water

(A)                                                           (B)

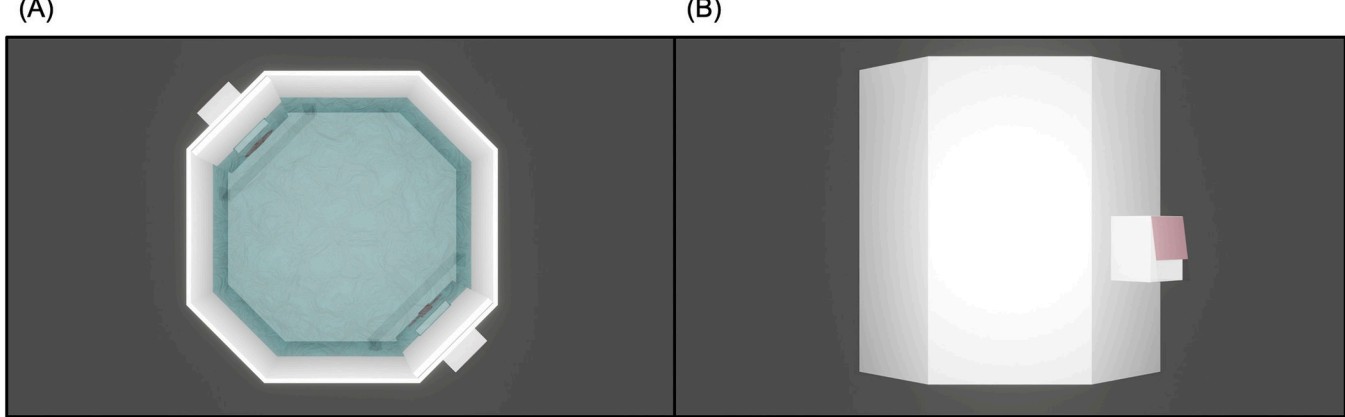

**Fig 2. Training arena.** In (A), the top view shows: The positioning of the training panels with the stimuli along the arena's diagonal; the transparent screens ahead of the stimuli; the position of the corridors towards the outside. In (B), the lateral view shows a detail of the open (rewarding) corridor with a flexible door at the end: fish could leave the training arena by swimming through the corridor and slightly pushing the door.

temperature was maintained constant at around 25±1˚C with the aid of a heater (NEWA Therm®, NEWA), while a filter (NEWA Duetto®, NEWA) ensured a good water quality (both the heater and the filter were not present during the experiment). Fish behavior was then recorded by using an overhead webcam (Life Cam Studio, Microsoft, USA).

**Procedure.** *Experiment 1* was designed in two phases: an exposure procedure to two-dimensional visual stimuli (exposure-phase), in the rearing tanks, and a learning procedure (training-phase), in the octagonal arena equipped for the visual discrimination learning task.

As mentioned above, the exposure-phase in the group within common rearing tanks lasted 50 days and continued within individual tanks during training pauses, until learning had been achieved. The individual exposure allowed for continuity between the familiar rearing and the novel tanks. Moreover, the fish could be easily recognized and continue its personal training. During these periods, fish were subjected to three experimental conditions: specifically, 14 fish were raised in the presence of full red disks, 14 fish in the presence of amputated red disks, and 14 fish in the simultaneous presence of both.

The training-phase consisted of daily sessions of 10 trials until the learning criterion, that is, at least 70% (or more) of correct choices per two consecutive daily sessions (binomial test for first choices p<0.03; binomial test for total choices p<0.001). This criterion is common to other similar discriminative learning experiments [18–25], and it is intended to maintain procedural continuity and to have the opportunity to perform behavioral comparisons. Before starting each experimental trial, the fish was brought from the comfortable zone surrounding the arena and gently transferred into a transparent glass cylinder (diameter: 6 cm, height: 8 cm, open at both ends), thus placed into the center of the arena. After 10 seconds (an acclimatization time after handling), the cylinder was gently lifted and removed, leaving the fish free to swim around and scan the two stimuli. In each trial, the choices made by the fish within the two corridors with the visual stimuli were scored, until the fish was able to go out into the surrounding zone or, in any case, for a 10-minute limit. A correction method was used [27]: specifically, the fish was allowed to change the wrong choices until it was able to exit or until the overall time for the trial elapsed. Inter-trial intervals, during which the fish was allowed to rest in the comfortable surrounding zone, were 6 minutes (complete reinforcement time, with the administration of a small amount of food), when the correct stimulus was identified on the first attempt, and 2 minutes (reduced reinforcement time), when the correct stimulus was identified after two or more attempts. Additionally, in the absence of attempts within the 10-minute limit (null trial), the fish was given a 5-minute break. Multiple choices for the correct corridor could occur, either because the fish explored the tunnel without getting out or did not exert enough strength to open the door. As individual data, the number of choices within the two corridors above the stimuli (i.e., the total number of choices per fish summed over the daily sessions of 10 trials) was used. An inter-observer reliability criterion [28] was applied in the re-coding of a subset of 10% of different videos (*p*<0.001, Pearson's correlation between the ratio calculated on the original coding and the *de novo* coding performed by an experimenter blind to the test condition of the fish).

Six experimental conditions were arranged: 1) fish raised with full red disk and rewarded with full red disk (N = 7; "congruence" condition); 2) fish raised with full red disk and rewarded with amputated red disk (N = 7; "wide-incongruence" condition; "wide-incongruence" is because this condition was not originally considered by Bateson & Chantrey [1]); 3) fish raised with amputated red disk and rewarded with amputated red disk (N = 7; "congruence" condition); 4) fish raised with amputated red disk and rewarded with full red disk (N = 7; "wide-incongruence" condition); 5) fish raised with a mixed condition, that is, full red disk and amputated red disk together, thus rewarded with full red disk (N = 7; "narrow-incongruence" condition; that is, "narrow" because similar to that proposed by Bateson & Chantrey

**Fig 3. Overview of Experiment 1.** Six experimental conditions were arranged: Reared full red disk-rewarded full red disk ("congruence"); reared full red disk-rewarded amputated red disk ("wide-incongruence"); reared amputated red disk-rewarded amputated red disk ("congruence"); reared amputated red disk-rewarded full red disk ("wide-incongruence"); reared mixed-rewarded full red disk ("narrow-incongruence"); reared mixed-rewarded amputated red disk ("narrow-incongruence"). The two narrow-incongruence conditions relate to the original experimental design by Bateson and Chantrey (1972).

[1]); 6) fish raised with a mixed condition, that is, full red disk and amputated red disk together, thus rewarded with amputated red disk (N = 7; "narrow-incongruence" condition). For a plain overview, see Fig 3.

## Experiment 2: Visual discrimination learning between a full red disk and a red cross

42 naïve *X. eiseni* were engaged in *Experiment 2*, which was performed by using the same apparatus and procedure employed in *Experiment 1*. Differences were essentially related to the exposure-phase: 14 fish were raised in the presence of full red disks (as in *Experiment 1*), 14 fish were raised in the presence of red crosses (differently from *Experiment 1*), and 14 fish in the simultaneous presence of both (full red disks and red crosses). The two-dimensional stimuli were full red disks ("scarlet", RGB: 255 red, 32 green, 0 blue; area: 4.52 cm$^2$, perimeter: 7.54 cm) and red crosses ("scarlet", RGB: 255 red, 32 green, 0 blue; area: 4.52 cm$^2$, perimeter: 11.4 cm); see Fig 1C. In this experiment, the magnitude of the stimuli was balanced following the area.

The two conditions of congruence and the four conditions of incongruence followed the rationale of *Experiment 1*, only by replacing the amputated red disk with the red cross. Thus, the six experimental conditions were: 1) fish raised with full red disk and rewarded with full red disk (N = 7; "congruence" condition); 2) fish raised with full red disk and rewarded with red cross (N = 7; "wide-incongruence" condition); 3) fish raised with red cross and rewarded

**Fig 4. Overview of Experiment 2.** Six experimental conditions were arranged: Reared full red disk-rewarded full red disk ("congruence"); reared full red disk-rewarded red cross ("wide-incongruence"); reared red cross-rewarded red cross ("congruence"); reared red cross-rewarded full red disk ("wide-incongruence"); reared mixed-rewarded full red disk ("narrow-incongruence"); reared mixed-rewarded red cross ("narrow-incongruence"). The two narrow-incongruence conditions relate to the original experimental design by Bateson and Chantrey (1972).

with red cross (N = 7; "congruence" condition); 4) fish raised with red cross and rewarded with full red disk (N = 7; "wide-incongruence" condition); 5) fish raised with a mixed condition, that is, full red disk and red cross together, thus rewarded with full red disk (N = 7; "narrow-incongruence" condition); 6) fish raised with a mixed condition, that is, full red disk and red cross together, thus rewarded with red cross (N = 7; "narrow-incongruence" condition). For a plain overview, see Fig 4.

## Experiment 3: Baseline definition

This control experiment was performed to assess both the presence of possible facilitatory effects of the exposure-phase on the congruence conditions and the presence of possible inhibitory effects of the exposure-phase on the incongruence conditions, comparing the results with those of *Experiment 1* and *2*. Moreover, it has been considered a feasible discriminative difficulty contingent on the stimulus per se (i.e., when the rewarded stimulus was the full red disk, the amputated red disk or the red cross) to verify if fish showed potential spontaneous preferences.

For these purposes, 28 naïve *X. eiseni* were engaged in the visual discrimination learning between a full red disk and an amputated red disk (N = 14), and between a full red disk and a red cross (N = 14), without being subjected to the exposure-phase. Half of each group was rewarded on the full red disk, while the other half on the amputated red disk (full red disk *versus* amputated red disk) or on the red cross (full red disk *versus* red cross). For a plain overview, see Fig 5. Results of the baseline were then compared with those obtained in *Experiment*

| Discrimination | Exposure-phase: Rearing stimuli | Training-phase: Rewarding stimuli |
|---|---|---|
| (full red disk, amputated red disk) | Absent | (full red disk) |
| (full red disk, amputated red disk) | Absent | (amputated red disk) |
| (full red disk, red cross) | Absent | (full red disk) |
| (full red disk, red cross) | Absent | (red cross) |

**Fig 5. Overview of Experiment 3.** Four experimental groups were arranged, two for the full red disk-amputated red disk discrimination (fish trained on the full red disk, fish trained on the amputated red disk), and two for the full red disk-red cross discrimination (fish trained on the full red disk, fish trained on the red cross), all in the absence of an exposure-phase.

*1* (*Visual discrimination learning between a full red disk and an amputated red disk*) and in *Experiment 2* (*Visual discrimination between a full red disk and a red cross*). The apparatus and the procedure were the same as those used in *Experiment 1* and *2*.

**Statistical analysis.** The considered variables were the mean number of trials (with 95% CI) to reach the learning criterion, the duration times (s) of all trials in the first three days of training (i.e., the minimum number of days in common among the experimental conditions), the duration times (s) of all trials until the learning criterion, the duration times (s) of all trials in the learning day (i.e., where fish achieved the learning criterion). The tests used to assess the homoscedasticity were the Levene Test of equality of error variances and Mauchly's Sphericity test. A Student's t-test and a one-way ANOVA were applied, to compare the congruence versus the incongruence conditions. To estimate the effect sizes, we reported partial eta-squared ($\eta_p^2$) as index for ANOVA and 95% Confidence Intervals for Student's t-test. Raw data were analyzed by using the IBM SPSS Statistics 20 software package.

The sample size in each experimental condition (N = 7) was similar to the one used in studies employing the same kind of tasks available in literature among fish species (for an example see [18–25]). Moreover, the number of animals observed was consistent with the ethical method of "Reduction", which allows considering the minimum number of animals to draw statistically valid results. The sample size has been approved by the Italian official bodies responsible for animal welfare, i.e., the Ethics Committee of the University of Trento and the National Ministry of Health, and it cannot exceed that declared in the approved official documents due to strict ethical reasons.

## Results

### Experiment 1: Visual discrimination learning between a full red disk and an amputated red disk

42 naïve *X. eiseni* were engaged for the visual discrimination learning between a full red disk and an amputated red disk, after being subjected to a phase of exposure to those stimuli, in

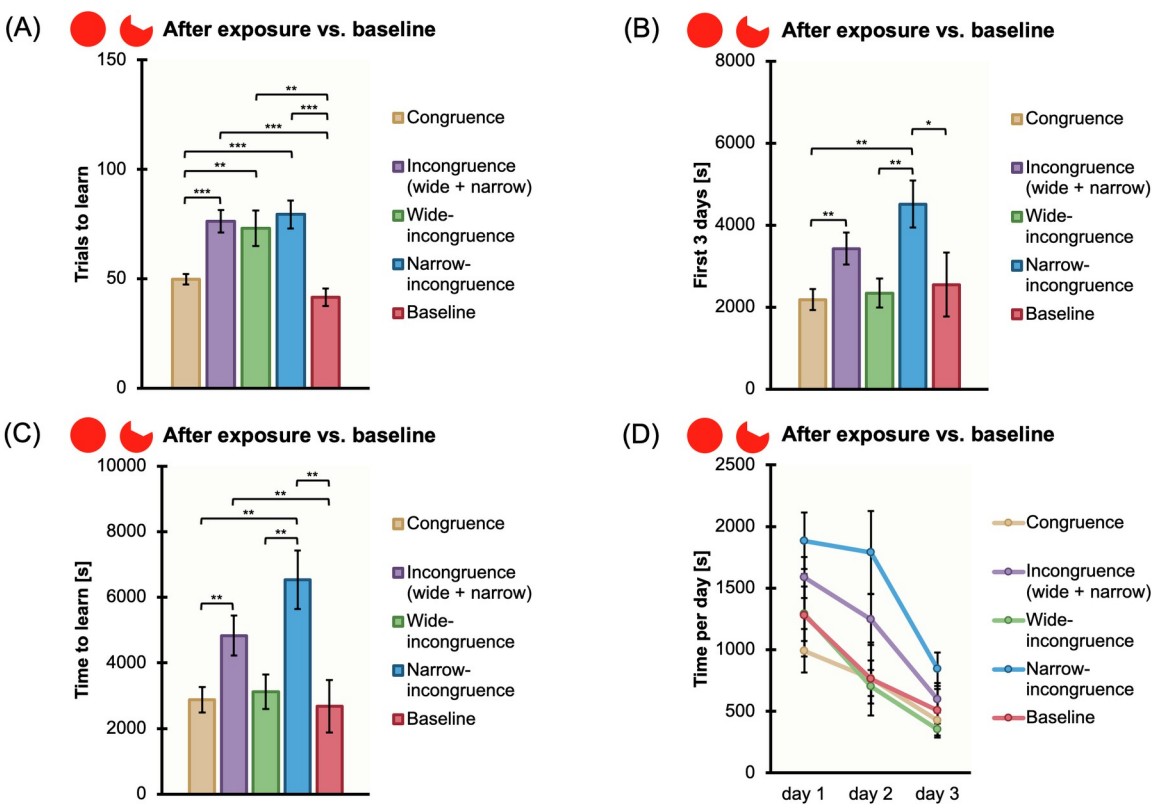

**Fig 6. Comparison among conditions and experiments.** The bar charts show all the comparisons among the congruence and incongruence conditions of Experiment 1 (full red disk vs. amputated red disk discrimination learning after exposure), and the comparison between Experiment 1 and the baseline of Experiment 3 (full red disk vs. amputated red disk discrimination learning without exposure), for the four variables measured: The mean number of trials to learn the discrimination, "Trials to learn" (A); the total amount of time spent, in seconds, during the experimental trials to exit the arena, considering the first 3 days of training, "First 3 days" (B); the total amount of time spent, in seconds, during the experimental trials to exit the arena, considering all the trials needed to meet the learning criterion, "Time to learn" (C); the time spent, in seconds, during the experimental trials to exit the arena, considering the first 3 days of training separately, "Time per day" (D). Mean ± SEM are shown; asterisks indicate statistically significant differences ($p \leq 0.05$).

congruence or incongruence (wide + narrow, wide, narrow) conditions. Results for Experiment 1 are reported in Fig 6A–6D.

**General analysis for congruence vs. incongruence (wide + narrow).** When comparing the two conditions of congruence, i.e., raised full red disk-rewarded full red disk, raised amputated red disk-rewarded amputated red disk *versus* the four conditions of incongruence (wide + narrow), i.e., raised full red disk-rewarded amputated red disk, raised amputated red disk-rewarded full red disk, raised mixed-rewarded full red disk, raised mixed-rewarded amputated red disk, we obtained the following results: there was a significant difference in the mean number of trials needed to reach the learning criterion (congruence: mean±SEM = 49.71 ± 2.43; wide + narrow-incongruence: mean±SEM = 76.25 ± 5.11, t(40) = -4.692, p $\leq$ 0.0001, 95% CI [-37.99, -15.07]), in the duration times (s) of the first three consecutive daily sessions (congruence: mean±SEM = 2186.029 ± 254.137; wide + narrow-incongruence: mean±SEM = 3429.935 ± 390.943, t(40) = -2.668, p = 0.011, 95% CI [-2186.41, -301.4]), and in the duration times (s) of all trials until the learning criterion (congruence: mean±SEM = 2878.23 ± 390.983; wide + narrow-incongruence: mean±SEM = 4833.314 ± 604.145, t(40) = -2.717, p = 0.01, 95% CI [-3409.68, -500.48]); there was not any difference in the duration times (s) of all trials in the

learning day (congruence: mean±SEM = 202.09 ± 32.34; wide + narrow-incongruence: mean ±SEM = 230.53 ± 49.85; t(40) = -0.38, p = 0.71).

The ANOVA with Time (the duration times in seconds of the first three consecutive daily sessions) as a within-subject factor and Congruency (the two conditions of congruence *versus* the four conditions of wide + narrow-incongruence) as a between-subject factor, revealed a significant main effect of Time (F(2,80) = 17.406, p ≤ 0.0001, $\eta_p^2$ = 0.303) and Congruency (F (1,40) = 4.538, p = 0.039, $\eta_p^2$ = 0.102), while the interaction between Time and Congruency was not significant (F(2,80) = 1.362, p = 0.262).

**Detailed analysis for congruence vs. wide-incongruence and congruence vs. narrow-incongruence.** When comparing the two conditions of congruence *versus* the two conditions of wide-incongruence, where fish were exposed with one of the two stimuli and rewarded with the other one, we obtained the following results: there was a significant difference in the mean number of trials needed to reach the learning criterion (congruence: mean ±SEM = 49.714 ± 2.426; wide-incongruence: mean±SEM = 73.07 ± 8.11; t(15) = -2.758, p = 0.01, 95% CI [-41.37, 5.34]); there was no significant difference in the duration times (s) of the first three consecutive daily sessions (congruence: mean±SEM = 2186.029 ± 254.137; wide-incongruence: mean±SEM = 2344.85 ± 357.02; t(26) = -0.362, p = 0.72), in the duration times (s) of all trials until the learning criterion (congruence: mean±SEM = 2878.23 ± 390.983; wide-incongruence: mean±SEM = 3125.99 ± 523.43; t(26) = -0.379, p = 0.71), and in the duration times (s) of all trials in the learning day (congruence: mean±SEM = 202.09 ± 32.34; wide-incongruence: mean±SEM = 163.26 ± 29.25; t(26) = 0.891, p = 0.38).

When comparing the two conditions of congruence *versus* the two conditions of narrow-incongruence where fish were exposed to both the stimuli, we obtained the following results: there was a significant difference in the mean number of trials needed to reach the learning criterion (congruence: mean±SEM = 49.714 ± 2.426; narrow-incongruence: mean ±SEM = 79.429 ± 6.407; t(26) = -4.337, p ≤ 0.0001, 95% CI [-44.19, -15.24]), in the duration times (s) of the first three consecutive daily sessions (congruence: mean ±SEM = 2186.029 ± 254.137; narrow-incongruence: mean±SEM = 4515.023 ± 571.188; t(26) = -3.725, p = 0.002, 95% CI [-3642.68, -1015.31]), and in the duration times (s) of all trials until the learning criterion (congruence: mean±SEM = 2878.23 ± 390.983; narrow-incongruence: mean±SEM = 6540.641 ± 890.887; t(26) = -3.764, p = 0.001, 95% CI [-5707.82, -1617.001]); there was not any difference in the duration times (s) of all trials in the learning day (congruence: mean±SEM = 202.09 ± 32.34; narrow-incongruence: mean±SEM = 297.79 ± 93.64; t(26) = -0.97, p = 0.34).

The ANOVA with Time (the duration times in seconds of the first three consecutive daily sessions) as a within-subject factor and Congruency (the two conditions of congruence *versus* the two conditions of narrow-incongruence, revealed a significant main effect of Time (F (2,52) = 13.045, p ≤ 0.0001, $\eta_p^2$ = 0.334) and Congruency (F(1,26) = 13.878, p = 0.001, $\eta_p^2$ = 0.348), while the interaction between Time and Congruency was not significant (F(2,52) = 1.836, p = 0.170).

**Detailed analysis for wide-incongruence vs. narrow-incongruence.** When comparing the two conditions of wide-incongruence *versus* the two conditions of narrow-incongruence, we obtained the following results: there was a significant difference in the duration times (s) of the first three consecutive daily sessions (wide-incongruence: mean±SEM = 2344.85 ± 357.02; narrow-incongruence: mean±SEM = 4515.02 ± 571.19; t(22) = 3.222, p = 0.004), and in the duration times (s) of all trials until the learning criterion (wide-incongruence: mean ±SEM = 3125.99 ± 523.43; narrow-incongruence: mean±SEM = 6540.64 ± 890.89; t(21) = 3.305, p = 0.003); there was not any difference in the mean number of trials needed to reach

the learning criterion (wide-incongruence: mean±SEM = 73.07 ± 8.11; incongruence: mean ±SEM = 79.43 ± 6.41; t(26) = 0.615, p = 0.54), and in the duration times (s) of all trials in the learning day (wide-incongruence: mean±SEM = 163.26 ± 29.25; narrow-incongruence: mean ±SEM = 297.79 ± 93.64; t(26) = 1.371, p = 0.18).

**Summary.** Results showed that fish took a higher number of trials to learn the discrimination between a full red disk and an amputated red disk under the incongruence conditions (novel stimulus) when compared to the congruence ones (familiar stimulus), even though already in the first three days of training there was an improvement of accuracy for both conditions (congruence and incongruence). Depending on the Congruency, a noteworthy effect of the duration times was observed. Specifically, the amount of time needed to complete each trial within the training session was greater in the conditions of incongruence with respect to the conditions of congruence, showing that fish took more time to discriminate between the full red disk and the amputated red disk in case of novelty, hence presumably highlighting the great complexity of such visual discrimination. Also, the amount of time in the first three days of training was higher under incongruence, where this effect was probably strictly related to the mixed exposure (narrow-incongruence). On the other hand, the absence of differences in the duration times in the learning day indicates that fish were equally adept at discriminating once they had reached the learning criterion, regardless of the experimental condition.

## Experiment 2: Visual discrimination learning between a full red disk and a red cross

42 naïve *X. eiseni* were engaged for the visual discrimination learning between a full red disk and a red cross, after being subjected to a phase of exposure to those stimuli, in congruence or incongruence (wide + narrow, wide, narrow) conditions. Results for Experiment 2 are reported in Fig 7A–7D.

**General analysis for congruence vs. incongruence (wide + narrow).** When comparing the conditions of congruence, i.e., raised full red disk-rewarded full red disk, raised red cross-rewarded red cross *versus* the four conditions of incongruence (wide + narrow), i.e., raised full red disk-rewarded red cross, raised red cross-rewarded full red disk, raised mixed-rewarded full red disk, raised mixed-rewarded red cross), there were the following results: there was a significant difference in the mean number of trials needed to reach the learning criterion (congruence: mean±SEM = 62.86 ± 7.59; wide + narrow-incongruence: mean±SEM = 92.50 ± 10.22, t(40) = -2.33, p = 0.025, 95% CI [-55.37, -3.91]); there was neither a significant difference in the duration times (s) of the first three consecutive daily sessions (congruence: mean ±SEM = 2562.97 ± 480.02; wide + narrow-incongruence: mean±SEM = 2501.65 ± 291.81; t (40) = 0.12, p = 0.91), nor in the duration times (s) of all trials until the learning criterion (congruence: mean±SEM = 3756.47 ± 855.55; wide + narrow-incongruence: mean±SEM = 3467.90 ± 418.23; t(40) = 0.34, p = 0.73); there was not any difference in the duration times (s) of all trials in the learning day (congruence: mean±SEM = 177.27 ± 46.59; wide + narrow-incongruence: mean±SEM = 114.65 ± 25.61; t(40) = 1.28, p = 0.21).

The ANOVA with Time (the duration times in seconds of the first three consecutive daily sessions) as a within-subject factor and Congruency (the two conditions of congruence *versus* the four conditions of wide + narrow-incongruence) as a between-subject factor, it revealed a significant main effect of Time (F(2,80) = 48.99, p $\leq$ 0.0001, $\eta_p^2$ = 0.55), while the Congruency (F(1,40) = 0.01, p = 0.91) and the interaction between Time and Congruency (F(2,80) = 0.05, p = 0.95) were not significant.

**Detailed analysis for congruence vs. wide-incongruence and congruence vs. narrow-incongruence.** When comparing the conditions of congruence *versus* the two conditions of

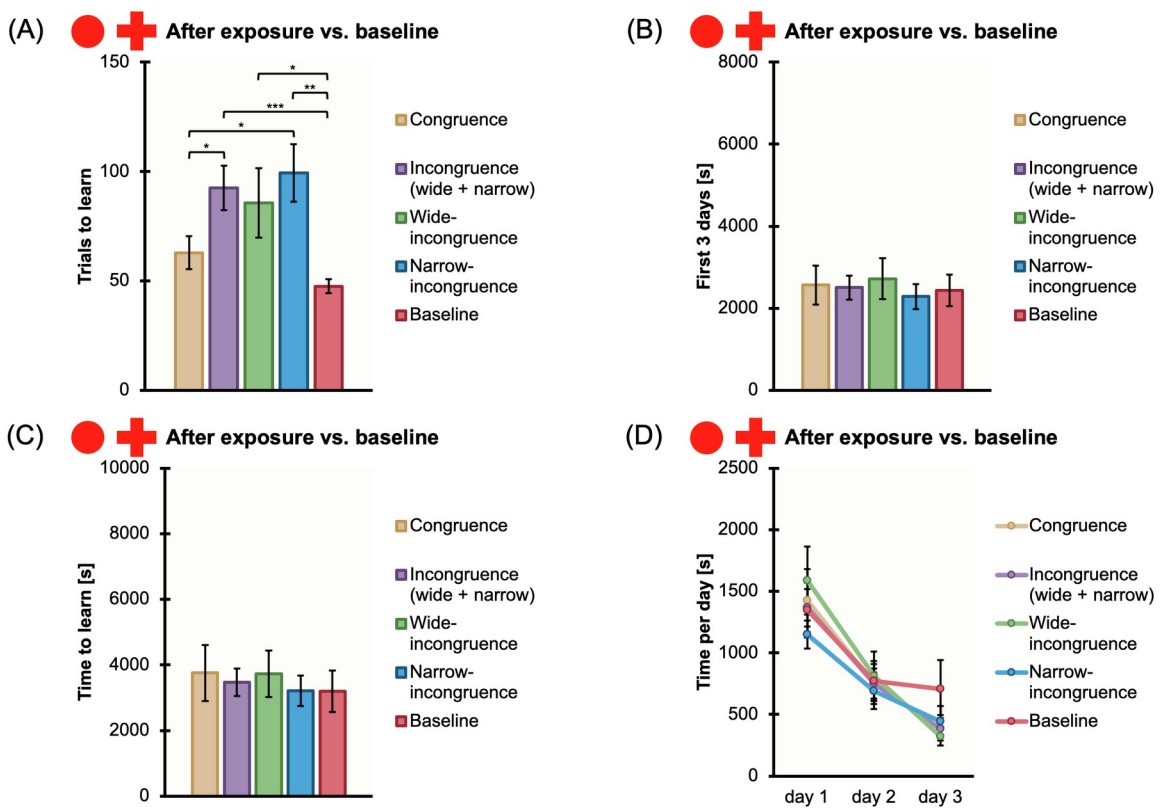

**Fig 7. Comparison among conditions and experiments.** The bar charts show all the comparisons among the congruence and incongruence conditions of Experiment 2 (full red disk vs. red cross discrimination learning after exposure), and the comparison between Experiment 2 and the baseline of Experiment 3 (full red disk vs. red cross discrimination learning without exposure), for the four variables measured: The mean number of trials to learn the discrimination, "Trials to learn" (A); the total amount of time spent, in seconds, during the experimental trials to exit the arena, considering the first 3 days of training, "First 3 days" (B); the total amount of time spent, in seconds, during the experimental trials to exit the arena, considering all the trials needed to meet the learning criterion, "Time to learn" (C); the time spent, in seconds, during the experimental trials to exit the arena, considering the first 3 days of training separately, "Time per day" (D).Mean ± SEM are shown; asterisks indicate statistically significant differences ($p \leq 0.05$).

wide-incongruence where fish were exposed with one of the two stimuli and rewarded with the other one, there were no statistically significant effects (mean number of trials needed to reach the learning criterion, congruence: mean±SEM = 62.86 ± 7.59; wide-incongruence: mean±SEM 85.71 ± 15.92; t(26) = -1.296, p = 0.21; duration times (s) of the first three consecutive daily sessions, congruence: mean±SEM = 2562.97 ± 480.02; wide-incongruence: mean ±SEM = 2719.28 ± 501.70; t(26) = -0.225, p = 0.82; duration times (s) of all trials until the learning criterion, congruence: mean±SEM = 3756.47 ± 855.55; wide-incongruence: mean ±SEM = 3726.21 ± 710.45; t(26) = 0.027, p = 0.98; duration times (s) of all trials in the learning day, congruence: mean±SEM = 177.27 ± 46.59; wide-incongruence: mean ±SEM = 130.06 ± 46.79; t(26) = 0.715, p = 0.48).

When comparing the conditions of congruence *versus* the two conditions of narrow-incongruence where fish were exposed to both the stimuli, we obtained the following results: there was a significant difference in the mean number of trials needed to reach the learning criterion (congruence: mean±SEM = 62.86 ± 7.59; narrow-incongruence: mean±SEM = 99.29 ± 13.15; t (26) = -2.40, p = 0.03, 95% CI [-68.03, -4.83]), while there was not a significant difference in the duration times (s) of the first three consecutive daily sessions (congruence: mean±SEM = 2562.97 ± 480.02; narrow-incongruence: mean±SEM = 2284.02 ± 307.77; t(26) = 0.49, p = 0.63),

in the duration times (s) of all trials until the learning criterion (congruence: mean±SEM = 3756 ± 855.55; narrow-incongruence: mean±SEM = 3209.60 ± 459.98; t(26) = 0.56, p = 0.58), and in the duration times (s) of all trials in the learning day (congruence: mean±SEM = 177.27 ± 46.59; narrow-incongruence: mean±SEM = 99.24 ± 22.33; t(26) = 1.51, p = 0.15).

The ANOVA with Time (the duration times in seconds of the first three consecutive daily sessions) as a within-subject factor and Congruency (the two conditions of congruence *versus* the two conditions of narrow-incongruence) as a between-subject factor, it revealed a significant main effect of Time (F(2,52) = 35.05, p ≤ 0.0001, $\eta_p^2$ = 0.57), while the Congruency (F (1,26) = 0.24, p = 0.63) and the interaction between Time and Congruency (F(2,52) = 1.26, p = 0.29) were not significant.

**Detailed analysis for wide-incongruence vs. narrow-incongruence.**   When comparing the two conditions of wide-incongruence *versus* the two conditions of narrow-incongruence, there were no statistically significant effects (mean number of trials needed to reach the learning criterion, wide-incongruence: mean±SEM = 85.71 ± 15.92; narrow-incongruence: mean ±SEM 99.29 ± 13.15; t(26) = 0.657, p = 0.52; duration times (s) of the first three consecutive daily sessions, wide-incongruence: mean±SEM = 2719.28 ± 501.70; narrow-incongruence: mean±SEM = 2284.02 ± 307.77; t(26) = -0.740, p = 0.47; duration times (s) of all trials until the learning criterion, wide-incongruence: mean±SEM = 3726.21 ± 710.45; narrow-incongruence: mean±SEM = 3209.60 ± 459.98; t(26) = -0.610, p = 0.55; duration times (s) of all trials in the learning day, wide-incongruence: mean±SEM = 130.06 ± 46.79; narrow-incongruence: mean ±SEM = 99.23 ± 22.33; t(26) = -0.594, p = 0.56).

**Summary.**   Results showed that fish took a higher number of trials to learn discriminating between a full red disk and a red cross under the incongruence conditions (novel stimulus) compared to the congruence ones (familiar stimulus), even though already in the first three days of training there was an improvement of accuracy for both conditions (congruence and incongruence). Unlike *Experiment 1*, in *Experiment 2* the effect of duration times was not significant. Specifically, the amount of time needed to complete each trial within the training session was similar in both conditions, showing that fish did not take longer to discriminate between the full red disk and the red cross, hence highlighting the greatest ease with which fish carried out such a visual discrimination. Furthermore, the duration times in the learning day were similar, indicating that fish were equally adept at discriminating once they had reached the learning criterion, regardless of the experimental conditions.

## Experiment 3: Baseline definition

28 naïve *X. eiseni* were engaged for the visual discrimination learning between a full red disk and an amputated red disk (N = 14), and between a full red disk and a red cross (N = 14). In this experiment, the phase of exposure to the stimuli was not expected. Results were compared with those of *Experiment 1* (full red disk—amputated red disk discrimination learning) and *Experiment 2* (full red disk—red cross discrimination learning).

**Full red disk—amputated red disk discrimination: general analysis for baseline vs. congruence and baseline vs. incongruence (wide + narrow).**   In the case of full red disk—amputated red disk discrimination, when comparing the baseline, i.e., no exposure-rewarded full red disk, no exposure-rewarded amputated red disk *versus* the two conditions of congruence of Experiment 1, i.e., raised full red disk-rewarded full red disk, raised amputated red disk-rewarded amputated red disk, there were no statistically significant effects (mean number of trials needed to reach the learning criterion, baseline: mean±SEM = 41.57 ± 3.99, congruence: mean±SEM = 49.71 ± 2.43, t(26) = 1.74, p = 0.096; duration times (s) of the first three consecutive daily sessions, baseline: mean±SEM = 2550.61 ± 778.45, congruence: mean

±SEM = 2186.03 ± 254.14, t(26) = -0.45, p = 0.66; duration times (s) of all trials until the learning criterion, baseline: mean±SEM = 2683.06 ± 797.85, congruence: mean ±SEM = 2878.23 ± 390.98, t(26) = 0.22, p = 0.83; duration times (s) of all trials in the learning day, baseline: mean±SEM = 368.03 ± 221.95, congruence: mean±SEM = 202.10 ± 32.34, t(26) = -0.74, p = 0.47).

On the other hand, when comparing the baseline, *versus* the four conditions of incongruence (wide + narrow) of Experiment 1, i.e., raised full red disk-rewarded amputated red disk, raised amputated red disk-rewarded full red disk, raised mixed-rewarded full red disk, raised mixed-rewarded amputated red, there was a significant difference in the mean number of trials needed to reach the learning criterion (baseline: mean±SEM = 41.57 ± 3.99, wide + narrow-incongruence: mean±SEM = 76.25 ± 5.11, t(40) = 5.35, p ≤ 0.0001, 95% CI [21.57, 47.79]), and in the duration times (s) of all trials until reaching the learning criterion (baseline: mean ±SEM = 2683.06 ± 797.85, wide + narrow-incongruence: mean±SEM = 4833.31 ± 604.15, t (40) = 2.099, p = 0.04, 95% CI [79.83, 4220.68]), while all the other variables were not significant (duration times (s) of the first three consecutive daily sessions, baseline: mean ±SEM = 2550.61 ± 778.45, wide + narrow-incongruence: mean±SEM = 3429.94 ± 390.94, t (40) = 1.13, p = 0.27; duration times (s) of all trials in the learning day, baseline: mean ±SEM = 368.03 ± 221.95, wide + narrow-incongruence: mean±SEM = 230.53 ± 49.85, t(40) = -0.60, p = 0.56).

**Full red disk—amputated red disk discrimination: detailed analysis for baseline vs. wide incongruence and baseline vs. narrow incongruence.** When comparing the baseline *versus* the two conditions of wide-incongruence of Experiment 1, there was a significant difference in the mean number of trials needed to reach the learning criterion (baseline: mean ±SEM = 41.57 ± 3.99, wide-incongruence: mean±SEM = 73.07 ± 8.11, t(19) = 3.483, p = 0.002, 95% CI [12.57, 50.43]), while all the other variables were not significant (duration times (s) of the first three consecutive daily sessions, baseline: mean±SEM = 2550.61 ± 778.45, wide-incongruence: mean±SEM = 2344.85 ± 357.02, t(26) = -0.240, p = 0.81; duration times (s) of all trials until reaching the learning criterion (baseline: mean±SEM = 2683.06 ± 797.85, wide-incongruence: mean±SEM = 3125.99 ± 523.43, t(26) = 0.464, p = 0.65; duration times (s) of all trials in the learning day, baseline: mean±SEM = 368.03 ± 221.95, wide-incongruence: mean ±SEM = 163.26 ± 29.25, t(26) = -0.915, p = 0.37).

When comparing the baseline *versus* the two conditions of narrow-incongruence of Experiment 1, there was a significant difference in the mean number of trials needed to reach the learning criterion (baseline: mean±SEM = 41.57 ± 3.99, narrow-incongruence: mean ±SEM = 79.43 ± 6.41, t(22) = 5.014, p ≤ 0.0001, 95% CI [22.19, 53.52]), in the duration times (s) of the first three consecutive daily sessions (baseline: mean±SEM = 2550.61 ± 778.45, narrow-incongruence: mean±SEM = 4515.02 ± 571.19, t(26) = 2.035, p = 0.05, 95% CI [-20.26, 3949.08]), and in the duration times (s) of all trials until reaching the learning criterion (baseline: mean±SEM = 2683.06 ± 797.85, narrow-incongruence: mean±SEM = 6540.64 ± 890.89, t (26) = 3.226, p = 0.003, 95% CI [1399.32, 6315.84]); there was not a significant effect in the duration times (s) of all trials in the learning day, (baseline: mean±SEM = 368.03 ± 221.95, narrow-incongruence: mean±SEM = 297.79 ± 93.64, t(26) = -0.292, p = 0.77).

**Full red disk—amputated red disk discrimination: analysis for spontaneous preference.** By comparing the fish rewarded with the full red disk and those rewarded with the amputated red disk in the baseline condition, there was a significant difference in the mean number of trials needed to reach the learning criterion (full red disk: mean ±SEM = 31.43 ± 1.43, amputated red disk: mean±SEM = 51.71 ± 5.73, t(12) = -3.44, p = 0.01, 95% CI [-34.35, -6.22]), while all the other variables were not significant (duration times (s) of the first three consecutive daily sessions, full red disk: mean±SEM = 2836.24 ± 1395.33,

amputated red disk: mean±SEM = 2264.98 ± 807.33, t(12) = 0.35, p = 0.73; duration times (s) of all trials until the learning criterion, full red disk: mean±SEM = 2845.01 ± 1393.11, amputated red disk: mean±SEM = 2521.11 ± 899.41, t(12) = 0.19, p = 0.85; duration times (s) of all trials in the learning day, full red disk: mean±SEM = 633.39 ± 435.08; amputated red disk: mean±SEM = 102.66 ± 26.56; t(12) = 1.22, p = 0.27).

**Summary.**   Results showed that when fish had to discriminate between a full red disk and an amputated red disk, under the congruence (exposure and training with familiar stimulus) or under the baseline (in the absence of exposure), there was not a facilitation effect of such exposure on the mean number of trials needed to reach the learning criterion. In the first three days of training, the improvement of performance was similar for both conditions (congruence and baseline), while there were no differences in the amount of time needed to complete each trial within the training session and in the learning day. On the other hand, when fish had to discriminate between a full red disk and an amputated red disk, under the incongruence (exposure and training with novel stimulus) or under the baseline (in the absence of exposure), an inhibitory effect of the unfamiliar exposure was evident in the mean number of trials needed to reach the learning criterion, as well as in the duration times of all trials until the learning criterion, both higher for the incongruence conditions with respect to the baseline. Moreover, also the amount of time in the first three days of training was higher under the incongruence conditions, where this effect was strictly related to the mixed exposure (narrow-incongruence).

There was a difference in the mean number of trials to learn depending on the rewarded stimulus: fish took a higher number of trials if trained on the amputated red disk than on the full red disk, while the other variables did not change in the two groups of fish.

**Full red disk—red cross discrimination: general analysis for baseline vs. congruence and baseline vs. incongruence (wide + narrow).**   In the case of full red disk—red cross discrimination, when comparing the baseline, i.e., no exposure-rewarded full red disk, no exposure–rewarded red cross *versus* the two conditions of congruence of Experiment 2, i.e., raised full red disk-rewarded full red disk, raised red cross-rewarded red cross, there were no statistically significant effects (mean number of trials needed to reach the learning criterion, baseline: mean±SEM = 47.57 ± 3.17, congruence: mean±SEM = 62.86 ± 7.59, t(26) = 1.86, p = 0.08; duration times (s) of the first three consecutive daily sessions, baseline: mean ±SEM = 2430.67 ± 382.90, congruence: mean±SEM = 2562.97 ± 480.02, t(26) = 0.22, p = 0.83; duration times (s) of all trials until the learning criterion, baseline: mean ±SEM = 3197.05 ± 630.17, congruence: mean±SEM = 3756.47 ± 855.55, t(26) = 0.53, p = 0.60; duration times (s) of all trials in the learning day, baseline: mean±SEM = 291.47 ± 82.51, congruence: mean±SEM = 177.27 ± 46.59, t(26) = -1.21, p = 0.24).

On the other hand, when comparing the baseline *versus* the four conditions of incongruence (wide + narrow) of Experiment 2, i.e., raised full red disk-rewarded red cross, raised red cross-rewarded full red disk, raised mixed-rewarded full red disk, raised mixed–rewarded red cross, there was a significant effect of the mean number of trials needed to reach the learning criterion (baseline: mean±SEM = 47.57 ± 3.17, wide + narrow-incongruence: mean ±SEM = 92.50 ± 10.22, t(40) = 4.2, p ≤ 0.0001, 95% CI [23.13, 66.72]), while all the other variables were not significant (duration times (s) of the first three consecutive daily sessions, baseline: mean±SEM = 2430.67 ± 382.90, wide + narrow-incongruence: mean ±SEM = 2501.65 ± 291.81, t(40) = 0.14, p = 0.87; duration times (s) of all trials until the learning criterion, baseline: mean±SEM = 3197.05 ± 630.17, wide + narrow-incongruence: mean ±SEM = 3467.90 ± 418.23 t(40) = 0.37, p = 0.72; duration times (s) of all trials in the learning day, baseline: mean±SEM = 291.47 ± 82.51, wide + narrow-incongruence: mean ±SEM = 114.67 ± 25.61, t(40) = -2.05, p = 0.06).

**Full red disk—red cross discrimination: detailed analysis for baseline vs. wide incongruence and baseline vs. narrow incongruence.** When comparing the baseline *versus* the two conditions of wide-incongruence of Experiment 2, there was a significant difference in the mean number of trials needed to reach the learning criterion (baseline: mean ±SEM = 47.57 ± 3.16, wide-incongruence: mean±SEM = 85.71 ± 15.92, t(14) = 2.350, p = 0.03, 95% CI [3.33, 72.96]), while all the other variables were not significant (duration times (s) of the first three consecutive daily sessions, baseline: mean±SEM = 2430.67 ± 382.90, wide-incongruence: mean±SEM = 2719.28 ± 501.70, t(26) = 0.457, p = 0.65; duration times (s) of all trials until reaching the learning criterion (baseline: mean±SEM = 3197.05 ± 630.17, wide-incongruence: mean±SEM = 3726.21 ± 710.45, t(26) = 0.557, p = 0.58; duration times (s) of all trials in the learning day, baseline: mean±SEM = 291.47 ± 82.51, wide-incongruence: mean ±SEM = 130.06 ± 46.79, t(26) = -1.702, p = 0.10).

When comparing the baseline *versus* the two conditions of narrow-incongruence of Experiment 2, there was a significant difference in the mean number of trials needed to reach the learning criterion (baseline: mean±SEM = 47.57 ± 3.16, narrow-incongruence: mean ±SEM = 99.29 ± 13.15, t(15) = 3.822, p = 0.002, 95% CI [22.79, 80.64]), and in the duration times (s) of all trials in the learning day, baseline: mean±SEM = 291.47 ± 82.51, narrow-incongruence: mean±SEM = 99.24 ± 22.33, t(15) = -2.249, p = 0.04, 95% CI [-364.53, -9.93]); there was not a significant effect in the duration times (s) of the first three consecutive daily sessions, baseline: mean±SEM = 2430.67 ± 382.90, narrow-incongruence: mean ±SEM = 2284.02 ± 307.77, t(26) = -0.299, p = 0.77, and in the duration times (s) of all trials until reaching the learning criterion (baseline: mean±SEM = 3197.05 ± 630.17, narrow-incongruence: mean±SEM = 3209.60 ± 459.98, t(26) = 0.016, p = 0.99.

**Full red disk—red cross discrimination: analysis for spontaneous preference.** By comparing the fish rewarded with the full red disk and those rewarded with the red cross in the baseline condition, there were no differences in all the considered variables (mean number of trials needed to reach the learning criterion, full red disk: mean±SEM = 46.43 ± 5.39, red cross: mean±SEM = 48.71 ± 3.73, t(12) = -0.35, p = 0.73; duration times (s) of the first three consecutive daily sessions, full red disk: mean±SEM = 2242.96 ± 426.41, red cross: mean ±SEM = 2618.37 ± 664.64, t(12) = -0.48, p = 0.64; duration times (s) of all trials until the learning criterion, full red disk: mean±SEM = 3436.39 ± 1077.01, red cross: mean ±SEM = 2957.72 ± 736.07, t(12) = 0.37, p = 0.72; duration times (s) of all trials in the learning day, full red disk: mean±SEM = 334.80 ± 142.05; red cross: mean±SEM = 248.14 ± 93.25; t(12) = 0.51, p = 0.62).

**Summary.** Results showed that when fish had to discriminate between a full red disk and a red cross, under the congruence (exposure and training with familiar stimulus) or under the baseline (in the absence of exposure), there was no facilitation effect of such exposure on the mean number of trials needed to reach the learning criterion. In the first three days of training, the improvement of performance was similar for both conditions (congruence and baseline), while there were no differences in the amount of time needed to complete each trial within the training session and in the learning day. On the other hand, when fish had to discriminate between a full red disk and a red cross, under the incongruence (exposure and training with novel stimulus) or under the baseline (in the absence of exposure), an inhibitory effect of the unfamiliar exposure was evident in the mean number of trials needed to reach the learning criterion, higher for the incongruence conditions with respect to the baseline.

There was no significant difference depending on the rewarded stimulus: fish trained on the full red disk and fish trained on the red cross achieved similar performance in all the variables under consideration.

All the results are summarized and presented in Table 1 and Figs 6 and 7.

**Table 1. Results of Experiment 1, 2, and 3: Résumé.** The table summarizes the main results (*p*-values) for the four variables measured in Experiments 1, 2, and 3, with respect to the following comparisons: Congruence vs. wide + narrow-incongruence (Co vs. Wi+Na); congruence vs. wide-incongruence (Co vs. Wi); congruence vs. narrow-Incongruence (Co vs. Na); wide-incongruence vs. narrow-incongruence (Wi vs. Na); baseline vs. congruence (Ba vs. Co); baseline vs. wide + narrow-incongruence (Ba vs. Wi+Na); baseline vs. wide-incongruence (Ba vs. Wi); baseline vs. narrow-incongruence (Ba vs. Na).

| Experiment | Comparison | Trials to learn | First 3 days [s] | Time to learn [s] | Learning day [s] |
|---|---|---|---|---|---|
| 1 | Co vs. Wi+Na | $p \leq 0.0001$ | $p = 0.01$ | $p = 0.01$ | $p = 0.71$ |
| 1 | Co vs. Wi | $p = 0.01$ | $p = 0.72$ | $p = 0.71$ | $p = 0.38$ |
| 1 | Co vs. Na | $p \leq 0.0001$ | $p = 0.002$ | $p = 0.001$ | $p = 0.34$ |
| 1 | Wi vs. Na | $p = 0.54$ | $p = 0.004$ | $p = 0.003$ | $p = 0.18$ |
| 1, 3 | Ba vs. Co | $p = 0.096$ | $p = 0.66$ | $p = 0.83$ | $p = 0.47$ |
| 1, 3 | Ba vs. Wi+Na | $p \leq 0.0001$ | $p = 0.27$ | $p = 0.004$ | $p = 0.56$ |
| 1, 3 | Ba vs. Wi | $p = 0.002$ | $p = 0.81$ | $p = 0.65$ | $p = 0.37$ |
| 1, 3 | Ba vs. Na | $p \leq 0.0001$ | $p = 0.05$ | $p = 0.003$ | $p = 0.77$ |
| 2 | Co vs. Wi+Na | $p = 0.025$ | $p = 0.91$ | $p = 0.73$ | $p = 0.21$ |
| 2 | Co vs. Wi | $p = 0.21$ | $p = 0.82$ | $p = 0.98$ | $p = 0.48$ |
| 2 | Co vs. Na | $p = 0.03$ | $p = 0.63$ | $p = 0.58$ | $p = 0.15$ |
| 2 | Wi vs. Na | $p = 0.52$ | $p = 0.47$ | $p = 0.55$ | $p = 0.56$ |
| 2, 3 | Ba vs. Co | $p = 0.08$ | $p = 0.22$ | $p = 0.60$ | $p = 0.24$ |
| 2, 3 | Ba vs. Wi+Na | $p \leq 0.0001$ | $p = 0.87$ | $p = 0.72$ | $p = 0.06$ |
| 2, 3 | Ba vs. Wi | $p = 0.03$ | $p = 0.65$ | $p = 0.58$ | $p = 0.10$ |
| 2, 3 | Ba vs. Na | $p = 0.002$ | $p = 0.77$ | $p = 0.99$ | $p = 0.04$ |

## Discussion

Our work aimed to investigate the "classifying-together" phenomenon in a species of fish (*X. eiseni*), a widely-known teleost in animal cognition studies (social laterality: [7,8]; spatial abilities: [9–16]; detour behavior: [17]; visual discrimination learning and optical illusions: [18–23]; numerical abilities: [24]). Given the inconsistency of previous evidence collected with other vertebrates in confirmation of this effect [4–6], we performed a series of experiments targeted to assess the existence of such a phenomenon in fish.

For this purpose, we set up two discrimination learning experiments after rearing different groups of fish with pairs of two-dimensional visual stimuli for 50 consecutive days. Moreover, to determine the existence of any spontaneous preference of fish towards one or more stimuli, we further added another discrimination learning experiment without providing any kind of exposure to those shapes. In all the experiments, the learning performance was observed by examining both the mean number of trials to learn the discrimination and the time spent during the experimental trials to exit the training arena.

The results of *Experiment 1* (full red disk—amputated red disk discrimination learning after exposure), indicated that fish took a higher number of trials to learn the visual discrimination after the incongruent exposure, both in wide (raised full red disk-rewarded amputated red disk, raised amputated red disk-rewarded full red disk, raised mixed-rewarded full red disk, raised mixed-rewarded amputated red disk) and narrow (raised mixed-rewarded full red disk, raised mixed-rewarded amputated red disk) condition, compared to the congruent exposure (raised full red disk-rewarded full red disk, raised amputated red disk-rewarded amputated red disk). Additionally, a noteworthy effect of the duration times emerged as a function of Congruency. Indeed, the amount of time needed to complete each trial within the training session was greater under incongruence than congruence conditions, showing that fish took longer to discriminate between the full red disk and the amputated red disk in case of novelty. This effect presumably emphasizes the great complexity of such visual discrimination for this

teleost species (*X. eiseni*), unlike other teleost, the zebrafish, which easily learn to discriminate between a full red disk and an amputated red disk but not between a full red disk and a red cross [25]. On the other hand, the absence of differences in the amount of time needed to complete each trial within the learning session indicates that fish were equally adept at discriminating once they had reached the learning criterion, regardless of the experimental condition. Moreover, already in the first three days of training, fish showed an improvement of accuracy for both conditions (congruence and incongruence).

Similarly to *Experiment* 1, in *Experiment 2* (full red disk—red cross discrimination learning after exposure), we found that fish took a higher number of trials to learn the visual discrimination task after the incongruent exposure, both in wide (raised full red disk-rewarded red cross, raised red cross-rewarded full red disk, raised mixed- rewarded full red disk, raised mixed-rewarded red cross) and narrow (raised mixed- rewarded full red disk, raised mixed-rewarded red cross) condition, if compared to the congruent exposure (raised full red disk-rewarded full red disk, raised red cross-rewarded red cross). Differently from *Experiment 1*, however, there was no effect of the duration times as a function of Congruency. In other words, the amount of time needed to exit the training arena was equal under congruence and incongruence conditions, showing that fish did not take more seconds to discriminate between the full red disk and the red cross in case of novelty. They indeed spent a similar amount of time to solve the visual discrimination task both in case of novelty and familiarity. In this case too, results did not indicate any difference in the amount of time needed to complete each trial within the learning session, thus suggesting that fish were equally adept to discriminate once they had reached the learning criterion, regardless of the experimental condition, highlighting the greatest ease with which fish carried out such visual discrimination. Moreover, already in the first three days of training, fish showed an improvement in accuracy for both conditions (congruence and incongruence).

In the *Baseline* (full red disk-amputated red disk, full red disk-red cross discrimination learning without exposure) we observed an effect of the visual shapes on the learning performance of *X. eiseni*. Specifically, the choices made by fish were not equally spread in the full red disk-amputated red disk discrimination, where a simpler discrimination learning emerged depending on the stimulus. In fact, the number of trials needed to reach the learning criterion was lower when the full red disk was the rewarded stimulus with respect to the amputated red disk. However, if there had been a spontaneous preference towards the full red disk, such behavior would also have occurred in the full red disk-red cross condition. In that case, by contrast, the learning performance of fish was similar regardless of the rewarded stimulus. This difference may reflect a tendency to perceive a symmetric two-dimensional shape, such as a full disk or a cross, instead of an asymmetric one, such as an amputated disk (see, for instance, [29]). On the other hand, as noted above, this ability is species-specific since *Danio rerio* showed difficulties in discriminating full symmetric stimuli [25].

In contrast to Gibson and colleagues [4,5] and Stewart and colleagues [6], our results suggest that in this species of fish, there was no facilitation effect related to the 50 days of exposure, neither in the full red disk-amputated red disk, nor in the full red disk-red cross discrimination. However, in both visual discrimination learning tasks a retardation effect related to the exposure was evident: under the conditions of incongruence, the discrimination required a greater number of trials to reach the learning criterion and, limited to the case of full red disk-amputated red disk, a greater amount of time to solve the discrimination task within the training sessions and in the first three consecutive days of learning. All these effects were emphasized in the case of narrow-incongruence, where fish underwent simultaneous exposure to two different visual shapes before being rewarded with one of them in the subsequent discrimination learning task.

Here we provide the first evidence that a species of fish (*X. eiseni*) is liable to the classifying-together phenomenon, thus corroborating the early observations by Beatson and Chantrey [1]. This phenomenon surfaces both for fish raised in the concurrent presence of two different stimuli and then rewarded with one of them (narrow-incongruence condition, as proposed by Bateson and Chantrey [1], and for those raised in the presence of one type of stimulus (e.g., the full red disk) and then rewarded with the other one (e.g., the amputated red disk) (wide-incongruence condition, not tested by Bateson and Chantrey [1] and considered here as a supplementary condition of incongruence). In the former case, the simultaneous exposure to two figural stimuli slows down the discrimination learning of each of the two stimuli individually, probably processing them as a single figural identity (classifying-together phenomenon in the literal sense). In the latter case, the exposure to a figural stimulus, then not rewarded during the discrimination training, would have slowed the learning due to the absence of familiarity or perceptual dissonance with respect to the rearing stimulus. Furthermore, an interesting effect arises with respect to the nature of the visual shapes: under incongruence, the full red disk-amputated red disk discrimination required not only a higher number of trials to learn, but also more time (in terms of latency) for making a choice towards one of the two stimuli, with respect to the full red disk-red cross discrimination. This unexpected evidence, in this animal species, may highlight an important difference in perceiving the similarity of the stimuli associated with the discrimination demand (the more similar they are–such as a disk and a disk missing only a small part—and the more difficult is to classify them apart; conversely, the less similar they are–such as a disk and a cross—and the easier is to classify them apart). On the other hand, there are also species-specific peculiarities that determine the discrimination complexity of the stimuli [25], as well as, as evidenced in other fish species, by modifying some physical attributes of the stimuli, such as color, size or density, other saliency-related effects could crop up and affect the discrimination learning performance [30,31].

The differences among species probably play a crucial role not only within the class of fishes, but also across the classes of vertebrates: different species populate peculiar ecological niches (aerial or terrestrial) and characteristic living environments, they may have adapted to "treat" visual elements in a different way for the purposes of learning and their relevance for survival, and precisely this species-specificity could perhaps partly explain the presence-absence of the Bateson effect. However, the non-replicability of the results with chicks could find its *raison d'être* perhaps in the different types of stimuli (cylinders of different colors vs. circles and squares) and procedure used in the experiments (imprinting on the stimuli before discrimination vs. absence of imprinting; pre-exposure times: 50 days vs. 30 minutes in the first two days of life), respectively, of Bateson and Chantrey [1] and Stewart and collaborators [6]. On the other hand, as also observed from the results reported here with the species of fish *X. eiseni*, it emerged that the type of stimuli used during training affects the discrimination difficulty of the same, which can then affect the value of the Bateson effect and the discrimination ability.

In conclusion, our study provides the first evidence about the consistency of the classifying-together effect in a fish species, further highlighting the impact of visual similarities on discrimination processes. These similarities seem to follow the rules of Gestalt psychology [32,33], in which visual objects are perceived as patterns or configurations rather than as individual components, further supporting the view that some basic perceptual principles are shared across human and non-human species, sharing the common living environment and adaptation demands from our planet, the Earth.

## Supporting information

**S1 Raw data.**
(XLSX)

## Acknowledgments

We wish to thank Ciro Petrone, Cristina Pascu, Michela Maffei, Erika Fontanari, and Giampaolo Morbioli for the animal care and welfare; Anastasia Morandi-Raikova for proofreading; Eva Sheardown and the other anonymous Reviewer for reading and commenting on the manuscript.

## Author Contributions

**Conceptualization:** Valeria Anna Sovrano, Davide Potrich, Veronica Mazza.

**Data curation:** Valeria Anna Sovrano, Greta Baratti, Davide Potrich, Tania Rosà.

**Formal analysis:** Valeria Anna Sovrano.

**Funding acquisition:** Valeria Anna Sovrano, Veronica Mazza.

**Investigation:** Greta Baratti, Davide Potrich.

**Methodology:** Greta Baratti, Davide Potrich, Tania Rosà.

**Project administration:** Valeria Anna Sovrano.

**Resources:** Valeria Anna Sovrano.

**Supervision:** Valeria Anna Sovrano.

**Visualization:** Greta Baratti, Davide Potrich, Veronica Mazza.

**Writing – original draft:** Valeria Anna Sovrano, Greta Baratti, Davide Potrich, Tania Rosà, Veronica Mazza.

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
