## [Decision Letter · Decision Letter 0]

7 Jun 2022

PONE-D-22-09159“Classifying-together” phenomenon in fish (Xenotoca eiseni): Simultaneous exposure to visual stimuli  impairs subsequent discrimination learningPLOS ONE

Dear Dr. Sovrano,

Thank you for submitting your manuscript to PLOS ONE. After careful consideration, we feel that it has merit but does not fully meet PLOS ONE’s publication criteria as it currently stands. Therefore, we invite you to submit a revised version of the manuscript that addresses the points raised during the review process.

We look forward to receiving your revised manuscript.

Kind regards,

Livia D'Angelo

Academic Editor

PLOS ONE

Journal Requirements:

5. Please include a separate caption for each figure in your manuscript. Please include a separate caption for each figure in your manuscript.

Reviewers' comments:

Reviewer's Responses to Questions

**Comments to the Author**

1. Is the manuscript technically sound, and do the data support the conclusions?

Reviewer #1: Yes

Reviewer #2: Yes

2. Has the statistical analysis been performed appropriately and rigorously? 

Reviewer #1: Yes

Reviewer #2: Yes

3. Have the authors made all data underlying the findings in their manuscript fully available?

Reviewer #1: Yes

Reviewer #2: Yes

4. Is the manuscript presented in an intelligible fashion and written in standard English?

Reviewer #1: No

Reviewer #2: No

5. Review Comments to the Author

Reviewer #1: This clever study investigates the “classifying-together” or “Bateson effect” in a teleost fish, Xenotoca eiseni. This is a well-written paper with persuasive findings. The multiple experiments and controls are well thought-out to rule out possible alternatives. The findings were novel and suitable for publication in PlosOne, thus I have very limited queries.

- The learning criterion was set at least 70% of correct trials. How was it established?

- The authors stated that “the exposure-phase lasted at least 50 days”; since the exposure phase in this kind of experiment is a relevant variable, I wonder whether the authors can specify the duration of the exposure. Moreover, I wonder whether a longer duration of the exposure-phase might influence the performance of the fish.

- In some conditions, the number of subjects seems small (n=7; e.g. lines 537-540). Can the authors justify their use of parametric statistics?

- Specify how the sample size was calculated.

Reviewer #2: The authors have conducted an extensive and clever set of experiments to investigate the phenomenon known as the "Bateson-effect" in a behavioral model that has been used in previous cognitive studies, the Redtail splitfin. The work done disentangles some of the uncertainty seen with studies of this phenomenon in other species, and also provides original evidence for this "classifying-together effect" in the teleost species, previously seen before only in higher order species. Also the interesting results found regarding differences in visual discrimination abilities for different shape stimuli presented under incongruence (easier discrimination of cross vs. circle) are discussed in detail leading in to discussion of potential differences seen in this body of work vs. the literature on the "Bateson-effect"/"classifying-together effect" in other species. The analysis and interpretation of the results are sound and thorough. Overall I think this research is insightful and contributes novel evidence in the field which will be of interest to a wide audience, therefore I recommend publication.

I would point out however, a few changes that I think need to be made before publication.

1. The main issue I have with the manuscript is that the general quality and clarity of writing needs improvement. Some of the phrasing is odd and can result in some confusion at what the author is trying to say. Also there are some typos (or maybe just odd word choice).

2. I would maybe choose to refer to the stimuli couples as "pairs" or "coupled" rather than 'a couple of' as you use in the manuscript. This is as this is more commonly used when referring to coupled stimuli in literature on visual discrimination.

3. There is a fair amount of methodical information at the end of the introduction which is not necessary to lead into your methods section. You really just want to give an overview of what the set of experiments is investigating rather than a whole lot of detail on the experiments as you are about to give that in the following methods section.

4. In line 128-129 you need a reference for the statement regarding male motivation to join female conspecifics.

5. When describing the stimuli pairs you say that they are balanced for perimeter (disc and amputated disc) and area (disc and cross) however both of these statements are not true. The values are different between the stimuli so you need to correct the statement.

6. The use of 'cozy' to describe the environment throughout is slightly anthropomorphizing, familiar or comfortable may be better.

7. There is a 10-second holding cylinder protocol at the start of the trials however it is not clear what this is for? Is it to view both stimuli? If this is the case can you be sure that the fish will have seen both in such a short period of time? If it is for another reason this needs to be stated.

8. The figures: Figures 6-9 would be better of being just two figure panels of 4 graphs (6a,7a,8a,9a then 6b,7b,8b,9b) to improve data visualization. There also don't seem to be any figure legends in the document I have been given. This makes the figures pretty useless as they should stand on their own without the manuscript text.

6. PLOS authors have the option to publish the peer review history of their article (what does this mean?). If published, this will include your full peer review and any attached files.

Reviewer #1: No

Reviewer #2: **Yes: **Eva Sheardown

---

## [Author Response · Author response to Decision Letter 0]

25 Jul 2022

Reviewer #1: This clever study investigates the “classifying-together” or “Bateson effect” in a teleost fish, Xenotoca eiseni. This is a well-written paper with persuasive findings. The multiple experiments and controls are well thought-out to rule out possible alternatives. The findings were novel and suitable for publication in PlosOne, thus I have very limited queries.

We were happy to read that our MS received positive feedback from the Reviewer. We thank the Reviewer for the useful comments and suggestions, which improved the quality of our work. 

- The learning criterion was set at least 70% of correct trials. How was it established?

The ≥ 70% criterion is based: (1) on the choice already made in similar discriminative learning experiments, to maintain procedural continuity and have the opportunity to draw comparisons (for example, see the most recent publication in this same Journal: Sovrano et al., 2022, “Visual discrimination and amodal completion in zebrafish”, PLoS One, 17(3): e0264127 https://doi.org/10.1371/journal. pone.0264127); (2) on the rationale that, having to choose between two options, the chance level is 50%, so 70% for one of the two options represents a consistent deviation from random behavioral choices. Moreover, the probability of performing 70% of correct choices maintained for two consecutive days was statistically different from a chance level: binomial test for first choices p < 0.03; binomial test for total choices p< 0.001. The joint use of these two dependent variables (first choices and total choices ≥70%) might be sensitive to different aspects of the animals’ performance (e.g. the total choices could be more sensitive to show within-session learning events.)

We now specify how the learning criterion was established in the text. 

- The authors stated that “the exposure-phase lasted at least 50 days”; since the exposure phase in this kind of experiment is a relevant variable, I wonder whether the authors can specify the duration of the exposure. Moreover, I wonder whether a longer duration of the exposure-phase might influence the performance of the fish.

All the fish were hosted in the rearing/exposure tank with the stimuli and other conspecifics for 50 days. Then, the fish continued to be exposed to those stimuli even during the rest time between each daily session of the discriminative learning task; in this case, the fish were kept separated in individual spaces, so that we could recognize each fish and continue its personal training.

We have hopefully improved the description of this section by better differentiating the exposure types (in group and individually) during the learning-phase, and by deleting the word "almost" to avoid confusion in the reader.

- In some conditions, the number of subjects seems small (n=7; e.g. lines 537-540). Can the authors justify their use of parametric statistics?

We considered 7 animals for each experimental condition in both discriminative learning experiments and in the baseline condition (not only for the baseline, lines 537-540). We have already indicated this in the Procedure section. The sample size satisfies the ethical principle of “Reduction”, according to which it is acceptable to use the smallest number of animals leading to statistically valid results.

Since the number of subjects used for each experimental condition has been extensively described in the Procedure, we have now deleted the single two values in the Results, to match the format of all the sub-paragraphs and to avoid confusion or doubts in the reader.

We used parametric statistics when the dependent variables could assume values from zero to infinite, such as the number of overall choices in each corridor, the number of trials to learn the discrimination, the time duration (in seconds) of each trial and their total amount. All the range values of these variables can be represented on a scale of greater complexity (such as the ratio scale, which also assumes an absolute zero).

- Specify how the sample size was calculated.

The sample size used here, in each experimental condition (N=7), was similar to the one used in studies with the same kind of tasks and power analyses already available in literature among fish species (for an example see [18-25]). The sample size was even approved by the Italian official bodies responsible for animal welfare, i.e., the Ethics Committee of the University of Trento and the National Ministry of Health, and it cannot exceed that declared in the approved official documents due to strict ethical reasons. We have now mentioned this crucial aspect in the text, and we thank the Reviewer for raising this issue.

Reviewer #2: The authors have conducted an extensive and clever set of experiments to investigate the phenomenon known as the "Bateson-effect" in a behavioral model that has been used in previous cognitive studies, the Redtail splitfin. The work done disentangles some of the uncertainty seen with studies of this phenomenon in other species, and also provides original evidence for this "classifying-together effect" in the teleost species, previously seen before only in higher order species. Also the interesting results found regarding differences in visual discrimination abilities for different shape stimuli presented under incongruence (easier discrimination of cross vs. circle) are discussed in detail leading in to discussion of potential differences seen in this body of work vs. the literature on the "Bateson-effect"/"classifying-together effect" in other species. The analysis and interpretation of the results are sound and thorough. Overall I think this research is insightful and contributes novel evidence in the field which will be of interest to a wide audience, therefore I recommend publication.

I would point out however, a few changes that I think need to be made before publication.

We are grateful to Eva Sheardown for her positive feedback. Thank you for the useful comments and suggestions, which improved the quality of our work. 

1. The main issue I have with the manuscript is that the general quality and clarity of writing needs improvement. Some of the phrasing is odd and can result in some confusion at what the author is trying to say. Also there are some typos (or maybe just odd word choice).

A native English speaker has read and carefully revised the manuscript to improve the general quality and clarity of writing.

2. I would maybe choose to refer to the stimuli couples as "pairs" or "coupled" rather than 'a couple of' as you use in the manuscript. This is as this is more commonly used when referring to coupled stimuli in literature on visual discrimination.

Done, thank you for the suggestion. 

3. There is a fair amount of methodical information at the end of the introduction which is not necessary to lead into your methods section. You really just want to give an overview of what the set of experiments is investigating rather than a whole lot of detail on the experiments as you are about to give that in the following methods section.

Following this suggestion, we deleted the methodological information at the end of the Introduction section. We left the description of the purpose of our experiments, briefly presenting the experimental conditions and formulating possible expectations, to progressively introduce the reader to the subsequent sections.

4. In line 128-129 you need a reference for the statement regarding male motivation to join female conspecifics.

We added the references as requested (lines 138-140).

5. When describing the stimuli pairs you say that they are balanced for perimeter (disc and amputated disc) and area (disc and cross) however both of these statements are not true. The values are different between the stimuli so you need to correct the statement.

Thanks for noticing the mistake. By making a copy-and-paste, we had left the area of the amputated disc, instead of correctly reporting the area of the cross. We have now reported the correct values. As regards to the perimeters, however, we have specified that we made them as balanced as possible (actually, with a slight difference between them).

6. The use of 'cozy' to describe the environment throughout is slightly anthropomorphizing, familiar or comfortable may be better.

We replaced the word “cozy” with “comfortable”.

7. There is a 10-second holding cylinder protocol at the start of the trials however it is not clear what this is for? Is it to view both stimuli? If this is the case can you be sure that the fish will have seen both in such a short period of time? If it is for another reason this needs to be stated.

The 10-second period is an acclimatization time after the manipulation, necessary before starting each trial, to reduce the normal physiological activation (e.g., after having caught the fish in the comfortable external area and placed it at the center of the experimental arena). 

The possibility to visually scan both stimuli occurred as soon as the fish was free to move within the arena (i.e., after lifting the transparent cylinder).

We provided this piece of information in the text (lines 231-235)

8. The figures: Figures 6-9 would be better of being just two figure panels of 4 graphs (6a,7a,8a,9a then 6b,7b,8b,9b) to improve data visualization. There also don't seem to be any figure legends in the document I have been given. This makes the figures pretty useless as they should stand on their own without the manuscript text.

We accepted this suggestion in the overall presentation of the figures.

We added the captions (which were missing in the previous version of the MS), adapting them to the new version of the figures.

---

## [Editor Report · Decision Letter 1]

27 Jul 2022

“Classifying-together” phenomenon in fish (Xenotoca eiseni): Simultaneous exposure to visual stimuli  impairs subsequent discrimination learning

PONE-D-22-09159R1

Dear Dr. Sovrano,

We’re pleased to inform you that your manuscript has been judged scientifically suitable for publication and will be formally accepted for publication once it meets all outstanding technical requirements.

Kind regards,

Livia D'Angelo

Academic Editor

PLOS ONE
---

## [Editor Report · Acceptance letter]

16 Aug 2022

PONE-D-22-09159R1 

“Classifying-together” phenomenon in fish *(Xenotoca eiseni):* Simultaneous exposure to visual stimuli impairs subsequent discrimination learning 

Dear Dr. Sovrano:

I'm pleased to inform you that your manuscript has been deemed suitable for publication in PLOS ONE. Congratulations! Your manuscript is now with our production department. 

Kind regards, 

on behalf of

Dr. Livia D'Angelo 

Academic Editor

PLOS ONE